# STABILIZING GRADIENT DESCENT VIA SECOND-ORDER CONTROL-THEORETIC DYNAMICS

## ABSTRACT

In this paper, we establish a fundamental connection between the stability of gradient descent dynamics and the curvature of the underlying loss landscape from a continuous-time perspective. We show that the sign of the real parts of the Hessian's eigenvalues directly governs the convergence behavior of gradient-based optimization. Through analytically tractable, low-dimensional toy examples, we demonstrate that gradient descent can diverge even in simple convex settings. To address this issue, we formulate gradient descent as a second-order dynamical system and introduce a controller that guarantees *locally asymptotic stability* by regulating the system's eigen-structure. Notably, we show that the proposed controller admits a variational interpretation and can be realized as a gradient guidance term augmenting the original gradient. Empirical results on numerical examples with various curvatures and learning rate validate our theoretical findings and demonstrate our proposed method improves both stability and convergence behaviors.

## 1 INTRODUCTION

Gradient descent (GD) discretely iterates $\boldsymbol{\theta}_{t+1} = \boldsymbol{\theta}_t - \eta \nabla(\boldsymbol{\theta_t})$ to optimize over the training loss $f$. GD based algorithm, such as stochastic gradient descent (SGD), is one of the fundamental optimization strategy for deep learning models. Existing works on the analysis of stability of GD have strong assumptions regarding the *convexity*, *sharpness* (the maximum eigenvalue of the Hessian of loss function), and *smoothness*.

Traditional framework (Nesterov, 2013) analyze gradient descent under the assumption of training loss is convex and $L$-smooth (i.e. a function $f$ is $L$-smooth if, $\forall \boldsymbol{\theta}, \boldsymbol{\theta}', ||\nabla f(\boldsymbol{\theta} - \nabla f(\boldsymbol{\theta}')|| \leq L||\boldsymbol{\theta} - \boldsymbol{\theta}'||,$). Under this framework, Ahn et al. (2022) proves that gradient descent converges only if the learning rate $\eta$ satisfies $\eta < \frac{2}{L}$, and diverge otherwise. On the other side, Cohen et al. (2021) empirically demonstrate that GD operates in the regime called the Edge of Stability (EoS), in which the sharpness hovers just above $\frac{2}{\eta}$, and the training loss behaves non-monotonically over short timescales, yet consistently decreases over long timescales. Meanwhile, Wu et al. (2018) prove GD is stable if $||H(\boldsymbol{\theta})||_{\boldsymbol{2}} \leq \frac{2}{\eta}$, which draws the connection between sharpness, as measured by the spectral norm of Hessian, and the convergence behaviors of GD.

Unfortunately, modern deep neural networks normally fail on these assumptions above, and exhibit non-convex or non-smoothness loss. It is still unclear for the convergence and stability behaviors in general loss function of neural network, yet very little is known on the theoretical side of solving the unstable convergence of general non-convex and non-smooth case.

In our paper, we consider gradient descent as a dynamical system (Zhu et al., 2018; Wu et al., 2018), and analyze the stability of gradient descent without constrains on curvature. Specifically, we start with the first order training dynamics of gradient descent derived from gradient flow. We transform this first order training dynamics into a system of second order differential equation using functional derivative. We theoretically analyze the stability under various curvature setting. Then we propose a controller term and prove our modified *asymptotically stabilize* gradient descent regardless of the curvature and smoothness on loss function. Based upon the theoretical proofs, we propose our controlled gradient descent in algorithm 1 and conduct empirical experiments on numerical examples with various curvatures and learning rates to prove the effectiveness of our method. The key contributions can be summarized as:

- **Stability of GD for various curvatures**: We show that the stability of gradient descent is related to the curvature of training loss. We formulate the training dynamics as a second-order ODE and prove the connections between curvature and sharpness $\lambda$, where $\lambda$ is the largest eigenvalue of Hessian at local minimum. We show that even if the learning rate $\eta$ is properly bounded by $\eta < \frac{2}{\lambda}$, gradient descent can still be unstable if the curvature of training loss is not strongly convex, see Table 1.

- **Asymptotically stabilized GD dynamics:** We design a controller term that regulating the eigen-structure of the dynamical system. We apply our controller term into the second-order ODE of GD training dynamics. We prove our controlled second-order ODE is asymptotically stable regardless of the curvature in Theorem 3.

- **Higher tolerance on learning rate:** Empirically, we observe our controlled gradient descent not only stabilize the training dynamics but also have higher tolerance on the learning rate than GD since our controller term alternate the eigen-structure of training Hessian, therefore increases the 2/sharpness threshold for a stable learning rate as in Figure 1.

- **Our Controlled gradient descent algorithm:** We convert our theoretical controller term on the second-order ODE into an extra term on $\frac{d\theta}{dt}$ in Equation 5 to modify the gradient update of GD. We formulate our controlled gradient descent algorithm in Algorithm 1.

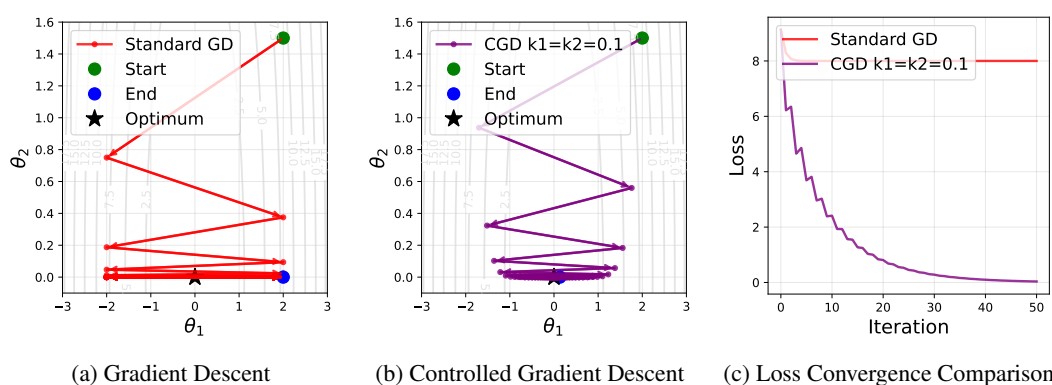

(a) Gradient Descent     (b) Controlled Gradient Descent     (c) Loss Convergence Comparison

Figure 1: Comparison between GD and controlled gradient descent for $L(\boldsymbol{\theta}) = 2\theta_1^2 + 0.5\theta_2^2$, with initial point $(\theta_1, \theta_2) = (2, 1.5)$ and learning rate $\eta = 0.5$. For our controlled term, we use $K_1 = k1I$ and $K_2 = k_2I$, where $k_1 = k_2 = 0.01$.

## 1.1 RELATED WORK

Many works observe the unstable convergence of GD. Jastrzebski et al. (2018); Xing et al. (2018) find that during GD, the training loss decreases non-monotonically, and than oscillating in between "valley walls". Lewkowycz et al. (2020) note that if the GD initialization has sharpness greater than $\frac{2}{\eta}$ during neural network training, GD will become initially destabilized, then exhibits "catapults" behavior into a flat region. Even though existing literature has observed the unstable training behaviors of GD and attempted to explain their underlying nature, it remains unclear how to resolve such instabilities. To date, no theoretically characterized algorithm exists that guarantees stabilized convergence of GD in general setting.

The Edge of Stability (EoS) is also a crucial view analyzing GD training dynamics. Cohen et al. (2021) formally point out the phenomenon of EoS where the largest eigenvalue of the Hessian hovers around threshold $\frac{2}{\eta}$. Furthermore, Long & Bartlett (2024); Dai et al. (2023) show that the Sharpness-Aware Minimization (SAM) stabilizes training dynamics and promote flatter minima with dynamically adjusted Hessian norm. Grimmer (2024) use periodically long step to speed up the rate of convergence for smooth and convex training loss. However, for other curvature cases, the stability analysis is insufficient and no existing method stabilize GD for training loss with general curvature.

Several works have considered GD training as a dynamical system. Zhu et al. (2018) formulate the training dynamics of stochastic gradient descent (SGD) as a stochastic differential equation

(SDE) to analyze the behavior of SGD on escaping from minima and its regularization effects. Wu et al. (2018) shows that learning rate and batch size play different roles in minima selection from the dynamical stability perspective. In this paper, we follow the dynamical system framework to analyze GD training as a second-order ODE, design controlling method accordingly, and propose our controlled gradient descent algorithm.

## 2 PRELIMINARY

In this section, we present necessary background knowledge on control theory of a dynamical system and gradient flow of gradient descent algorithm. We define the equilibrium of a dynamical system:

**Definition 1** (Equilibrium Point). *(Khalil, 2002) Let $\frac{d\boldsymbol{x}}{dt} = f(\boldsymbol{x})$ be a nonlinear dynamical system. A point $\boldsymbol{x}^*$ is called an equilibrium point if $f(\boldsymbol{x}^*) = \boldsymbol{0}$.*

For a dynamical system with a desired equilibrium, we can characterize its stability toward this equilibrium. Specifically, we define the following types of stability:

**Definition 2.** *(Glendinning, 1994) Let $\boldsymbol{x}(t)$ be a solution to a dynamical system $\frac{d\boldsymbol{x}}{dt} = f(\boldsymbol{x})$ with equilibrium point $\boldsymbol{x}^*$. This system is*

- ***Locally Lyapunov stable*** *if for every $\varepsilon > 0$, there exists a $\delta > 0$, such that when $\|\boldsymbol{x}(0) - \boldsymbol{x}^*\| < \delta$, $\|\boldsymbol{x}(t) - \boldsymbol{x}^*\| < \varepsilon$ for all $t \geq 0$.*

- ***Locally asymptotically stable*** *if it is Lyapunov stable and in addition, there exists a $\delta > 0$, such that when $\|\boldsymbol{x}(0) - \boldsymbol{x}^*\| < \delta$, $\lim_{t \to \infty} \boldsymbol{x}(t) = \boldsymbol{x}^*$.*

- ***Unstable*** *otherwise.*

**Remark 1.** *(Ak Gümüş, 2014) For locally (asymptotic) stability, solutions must approach an equilibrium point under initial conditions close to the equilibrium point. In globally (asymptotic) stability, solutions must approach to an equilibrium point under all initial conditions.*

When dealing with a complicated non-linear dynamical system, we can use the local linearization method to analyze its local stability around equilibrium with the following theorem:

**Theorem 1.** *[Local Stability via Linearization] (Perko, 2008) Let $\frac{d\boldsymbol{x}}{dt} = \boldsymbol{f}(\boldsymbol{x})$ be a continuously differentiable vector field, and let $\boldsymbol{x}^*$ be an equilibrium point. Consider the Jacobian matrix $\boldsymbol{J} = D\boldsymbol{f}(\boldsymbol{x}^*)$. Then the local stability of $\boldsymbol{x}^*$ is characterized as follows:*

1. *If all eigenvalues satisfy $Re(\lambda_i) \leq 0$ and every eigenvalue $\lambda_i$ with $Re(\lambda_i) = 0$ must have jordan blocks of size $1 \times 1$, then the system is said to be **locally Lyapunov stable**.*

2. *If all eigenvalues of $\boldsymbol{J}$ have strictly negative real parts, i.e., $Re(\lambda_i) < 0$ for all $i$, then $\boldsymbol{x}^*$ is **locally asymptotically stable**.*

3. *If at least one eigenvalue satisfies $Re(\lambda_i) > 0$, then $\boldsymbol{x}^*$ is **unstable**.*

**Definition 3** (Gradient Flow). *(Poliak, 1987) Let $L : \mathbb{R}^n \to \mathbb{R}$ be a continuously differentiable function. The* gradient flow *associated with $L$ is the solution to the first-order ODE:*

$$\frac{d\boldsymbol{\theta}}{dt} = -\nabla L(\boldsymbol{\theta}), \tag{1}$$

*with initial condition $\boldsymbol{\theta}(0) = \boldsymbol{\theta}_0$*

## 3 GRADIENT FLOW AND SECOND-ORDER DYNAMICS

We begin with the standard gradient flow dynamics (Eq. 1) to model gradient descent in continuous time setting, where $\boldsymbol{\theta} \in \mathbb{R}^d$ represents the parameters of neural networks and $L(\boldsymbol{\theta}) \in \mathbb{R}^d \to \mathbb{R}$ is the corresponding loss function.

Taking the time derivative of both sides yields the second-order dynamics:

$$\frac{d^2\boldsymbol{\theta}}{dt^2} = -\frac{d}{dt}\nabla L(\boldsymbol{\theta}) = -\nabla^2 L(\boldsymbol{\theta}) \cdot \frac{d\boldsymbol{\theta}}{dt}.$$

Hence, the second-order ODE is:

$$\frac{d^2\boldsymbol{\theta}}{dt^2} = -H(\boldsymbol{\theta}) \cdot \frac{d\boldsymbol{\theta}}{dt} \tag{2}$$

where $H(\boldsymbol{\theta}) = \nabla_L^2(\boldsymbol{\theta})$ is the Hessian matrix of the loss function.

## 4 Reformulation and Local Linearization of Second-Order ODE

In this section, we transform our second order ODE into a first order one and analyze the stability of second order ODE derived from the gradient descent algorithm by cases dividing upon the curvature of the loss function. We utilize Theorem 1 and relate the eigenvalues of the Hessian to the curvature of the training loss, then derive the stability analysis based on different settings.

### 4.1 First-Order System Reformulation

To express our second-order ODE as a first-order system, define the auxiliary variable: $\boldsymbol{x} = \frac{d\boldsymbol{\theta}}{dt}$. Then define the state vector: $\boldsymbol{z} = \begin{bmatrix} \boldsymbol{\theta} \\ \boldsymbol{x} \end{bmatrix} \in \mathbb{R}^{2n}$, and define the dynamics:

$$\frac{d\boldsymbol{z}}{dt} = f(\boldsymbol{z}) = \begin{bmatrix} \frac{d\boldsymbol{\theta}}{dt} \\ \frac{d\boldsymbol{x}}{dt} \end{bmatrix} = \begin{bmatrix} \boldsymbol{x} \\ -H(\boldsymbol{\theta}) \cdot \boldsymbol{x} \end{bmatrix}. \tag{3}$$

We now compute the Jacobian $J(\boldsymbol{z}) = \frac{\partial f}{\partial \boldsymbol{z}}$, which has the block structure $J(\boldsymbol{z}) = \begin{bmatrix} \frac{\partial f_1}{\partial \boldsymbol{\theta}} & \frac{\partial f_1}{\partial \boldsymbol{x}} \\ \frac{\partial f_2}{\partial \boldsymbol{\theta}} & \frac{\partial f_2}{\partial \boldsymbol{x}} \end{bmatrix}$, where: $f_1(\boldsymbol{\theta}, \boldsymbol{x}) = \boldsymbol{x}$, $f_2(\boldsymbol{\theta}, \boldsymbol{x}) = -H(\boldsymbol{\theta})\boldsymbol{x}$. Thus, the full Jacobian is:

$$J(\boldsymbol{z}) = \begin{bmatrix} 0 & I \\ -\sum_{i=1}^{n} \boldsymbol{x}_i \frac{\partial H(\boldsymbol{\theta})}{\partial \boldsymbol{\theta}_i} & -H(\boldsymbol{\theta}) \end{bmatrix}.$$

### 4.2 Local Stability at Equilibrium

During the training process of gradient descent, our goal is to reach the condition that $\boldsymbol{\theta} = \boldsymbol{\theta}^*$ and $\boldsymbol{x} = 0$, which is when $\boldsymbol{z}^* = \begin{bmatrix} \boldsymbol{\theta}^* \\ 0 \end{bmatrix}$. This goal is an equilibrium point, since $\boldsymbol{z}^*$ satisfies: $f(\boldsymbol{z}^*) = 0$

At equilibrium $\boldsymbol{z}^*$, we have: $J(\boldsymbol{z}^*) = \begin{bmatrix} 0 & I \\ 0 & -H(\boldsymbol{\theta}^*) \end{bmatrix}$. To investigate the stability, consider the characteristic equation of $J(\boldsymbol{z}^*)$. Let $\lambda \in \mathbb{C}$ be an eigenvalue, and we write $H = H(\boldsymbol{\theta}^*)$ for simplicity. Then:

$$\det \left( \begin{bmatrix} 0 & I \\ 0 & -H \end{bmatrix} - \lambda I \right) = \det \left( \begin{bmatrix} -\lambda I & I \\ 0 & -H - \lambda I \end{bmatrix} \right) = \det(\lambda^2 I + \lambda H) = \prod_{i=1}^{n} \lambda(\lambda + \lambda_i),$$

where $\lambda_i > 0$ are the eigenvalues of $H$. Therefore, the property of the eigenvalues of the Jacobian matrix is determined by the eigenvalues of the training loss Hessian.

**Theorem 2.** *The first order dynamic $\frac{d\boldsymbol{z}}{dt} = f(z)$ is:*

- *locally Lyapunov stable if the loss function $L$ is strongly convex (proof in Section 4.2.1)*

- *unstable if the loss function $L$ is convex but not strongly convex (proof in Section 4.2.2)*

- *unstable if the loss function $L$ is convex but not strongly concave (proof in Section 4.2.3)*

### 4.2.1 Strongly Convex Case

**Lemma 1.** *(Proof in Appendix A) [Strong Convexity and Positive Definiteness of the Hessian] Let $L : \mathbb{R}^n \to \mathbb{R}$ be a twice continuously differentiable function. Then, $L$ is strongly convex if and only if there exists a constant $m > 0$ such that the Hessian satisfies*

$$\nabla^2 L(\boldsymbol{\theta}) \succeq mI \quad \text{for all } \boldsymbol{\theta} \in \mathbb{R}^n.$$

*Equivalently, $L$ is strongly convex if and only if $\nabla^2 L(\boldsymbol{\theta})$ is positive definite for all $\boldsymbol{\theta} \in \mathbb{R}^n$.*

Table 1: Comparison of curvature assumption and stability between original gradient descent and our controlled gradient descent

| | Original Gradient Descent | | | Our Controlled Gradient Descent |
|---|---|---|---|---|
| Curvature Assumption | Strongly Convex | Convex (not strongly) | Concave | **None** |
| Stable | ✓ | ✗ | ✗ | ✓ |
| Asymptotically Stable | ✗ | ✗ | ✗ | ✓ |

Suppose $L$ is *strongly convex*. Then there exists $m > 0$ such that $H \succeq mI$, implying that $H$ is symmetric positive definite. The characteristic polynomial of $J$ is $\prod_{i=1}^n \lambda(\lambda + \lambda_i)$, where $\lambda_i > 0$ are the eigenvalues of $H$. This yields $n$ eigenvalues at $\lambda = 0$, and $n$ eigenvalues at $\lambda = -\lambda_i < 0$.

To study the Jordan form of $J$, consider the eigenspace of $\lambda = 0$:

$$J\boldsymbol{v} = 0 \quad \Rightarrow \quad \begin{bmatrix} 0 & I \\ 0 & -H \end{bmatrix} \begin{bmatrix} \boldsymbol{v}_1 \\ \boldsymbol{v}_2 \end{bmatrix} = \begin{bmatrix} \boldsymbol{v}_2 \\ -H\boldsymbol{v}_2 \end{bmatrix} = \boldsymbol{0}.$$

This implies $\boldsymbol{v}_2 = 0$, and thus $\boldsymbol{v}_1 \in \mathbb{R}^n$ is arbitrary. Therefore, the nullspace has dimension $n$, and the geometric multiplicity of the zero eigenvalue is $n$, equal to its algebraic multiplicity. Hence, all Jordan blocks associated with $\lambda = 0$ are $1 \times 1$. Therefore by Theorem 1, the first order dynamic $\frac{d\boldsymbol{z}}{dt} = f(z)$ is locally Lyapunov stable if the loss function $L$ is strongly convex.

### 4.2.2 Convex but Not Strongly Convex Case

**Lemma 2.** *(Proof in Appendix B) [Convexity and Positive Semidefiniteness of the Hessian] Let $L : \mathbb{R}^n \to \mathbb{R}$ be a twice continuously differentiable function. Then $L$ is convex if and only if the Hessian satisfies*

$$\nabla^2 L(\boldsymbol{\theta}) \succeq 0 \quad \text{for all } \boldsymbol{\theta} \in \mathbb{R}^n.$$

*Equivalently, $L$ is convex if and only if the Hessian is positive semidefinite at all points.*

Now assume $L$ is convex but not strongly convex. Then $H \succeq 0$, but $H$ is only positive semidefinite, meaning it has at least one eigenvalue equal to zero. Suppose $\lambda_1 = 0$ and $\boldsymbol{v}_1$ is the corresponding eigenvector. Then the characteristic polynomial becomes $\prod_{i=1}^n \lambda(\lambda + \lambda_i)$, with at least one repeated root at $\lambda = 0$. The algebraic multiplicity of $\lambda = 0$ is greater than $n$. Therefore, the geometric multiplicity is strictly less than the algebraic multiplicity. This implies that the Jordan block associated with $\lambda = 0$ has size strictly greater than $1 \times 1$.

Although all eigenvalues satisfy $\text{Re}(\lambda) \leq 0$, the existence of a Jordan block larger than $1 \times 1$ for an eigenvalue on the imaginary axis (specifically, $\lambda = 0$) violates the condition for marginal stability, resulting in solutions that grow linearly over time. Therefore by Theorem 1, the first order dynamic $\frac{d\boldsymbol{z}}{dt} = f(z)$ is unstable if the loss function $L$ convex but not strongly convex.

### 4.2.3 Concave Case

**Lemma 3.** *(Proof in Appendix C) [Concavity and Negative Semidefinite Hessian] Let $L : \mathbb{R}^n \to \mathbb{R}$ be twice continuously differentiable. Then $L$ is concave if and only if the Hessian satisfies*

$$\nabla^2 L(\boldsymbol{\theta}) \preceq 0 \text{ for all } \boldsymbol{\theta} \in \mathbb{R}^n,$$

*Equivalently, $L$ is concave if and only if the Hessian is negative semidefinite at all points.*

At a critical point, the characteristic polynomial is $\prod_{i=1}^n \lambda(\lambda + \lambda_i)$. Because $\lambda_i \leq 0$, the spectrum of $J$ is contained in $\{0\} \cup [0, \infty)$:

$$\text{spec}(J) = \underbrace{\{0, \dots, 0\}}_{n \text{ times}} \cup \{-\lambda_i(H)\}_{i=1}^n \subseteq \{0\} \cup (0, \infty).$$

Therefore by Theorem 1, the dynamical system of GD is unstable if the loss function $L$ is concave.

## 5 Controlling and Stabilizing Second Order ODE

In this section, we build from the training dynamic of gradient descent in Equation 2 and propose a controller term to stabilize gradient descent regardless of the curvature of the training objective. Specifically, we formulate a controller function $\boldsymbol{u}$ and transform our original training dynamic into

$$\frac{d^2\boldsymbol{\theta}'}{dt^2} = \frac{d^2\boldsymbol{\theta}}{dt^2} + \boldsymbol{u} = -H(\boldsymbol{\theta}) \cdot \frac{d\boldsymbol{\theta}}{dt} + \boldsymbol{u} \tag{4}$$

**Definition 4.** *Let the controller term $\boldsymbol{u} = -K_1\boldsymbol{\theta} - K_2\frac{d\boldsymbol{\theta}}{dt}$, where $K_1, K_2$ are $R^{d\times d}$ matrix, $K_1 \succ 0$ and $H(\boldsymbol{\theta}) + K_2 \succ 0$.*

**Remark 2.** *Empirically, $\boldsymbol{u}$ can be selected by choosing $K_1 \succ 0$ by letting $K_1 = \mu I$ for some $\mu > 0$ and choosing $K_2$ such that $K_2 \succ -H(\boldsymbol{\theta})$ for all $\boldsymbol{\theta}$.*

Substitute the controller term $\boldsymbol{u}$ in Definition 4 we get

$$\frac{d^2\boldsymbol{\theta}'}{dt^2} = -(H(\boldsymbol{\theta}) + K_2) \cdot \frac{d\boldsymbol{\theta}}{dt} - K_1\boldsymbol{\theta}.$$

We define $\dot{\boldsymbol{\theta}} = \frac{d\boldsymbol{\theta}}{dt}$ convert this second order ODE into first order ODE such that

$$\frac{d}{dt}\begin{bmatrix}\boldsymbol{\theta}\\\dot{\boldsymbol{\theta}}\end{bmatrix} = \underbrace{\begin{bmatrix}0 & I\\-K_1 & -(H(\boldsymbol{\theta}) + K_2)\end{bmatrix}}_{J(\boldsymbol{\theta})}\begin{bmatrix}\boldsymbol{\theta}\\\dot{\boldsymbol{\theta}}\end{bmatrix}.$$

We denote $J(\boldsymbol{\theta})$ as the Jacobian matrix of the system. By Theorem 1, system 4 is locally asymptotically stable around an equilibrium $\begin{bmatrix}\boldsymbol{\theta}\\\dot{\boldsymbol{\theta}}\end{bmatrix} = \begin{bmatrix}\boldsymbol{\theta}^*\\0\end{bmatrix}$ if all eigenvalues of $J(\boldsymbol{\theta}^*)$ have strictly negative real parts.

**Lemma 4.** *(Tisseur & Meerbergen, 2001) Let $Q(\lambda) = \lambda^2 M + \lambda C + K$ be a matrix-valued quadratic polynomial. Suppose $M \succ 0$, $C \succ 0$, and $K \succ 0$, then all eigenvalues $\lambda$ of $Q(\lambda)$ have strictly negative real parts.*

The characteristic equation for seeking the eigenvalues $\lambda$ of the Jacobian equation above is $\det(\lambda I - J) = 0$. this leads to a matrix-valued quadratic eigenvalue problem (QEP):

$$Q(\lambda) = \lambda^2 I + \lambda(H + K_2) + K_1.$$

**Theorem 3.** *The controlled second order system of gradient descent in Equation 4 is locally asymptotically stable*

*Proof.* In our system:

$$M = I, \quad C = H + K_2, \quad K = K_1.$$

Therefore by Lemma 4, all the eigenvalues $\lambda$ have negative real parts. As a result, by Theorem 1, system 4 is locally asymptotically stable, whereas the original system is Lyapnuov stable only under strongly convex loss. $\qquad\square$

Notice that the locally asymptotically stable guarantee for our controlled second order system is regardless of its curvature. We present a comparison of the theoretical stability guarantees for training dynamics between GD under various curvatures and our controlled gradient descent in general case in Table 1. Our controlled gradient descent not only relaxes the constrain on curvature, but also achieves better stability than GD for all curvature settings.

## 6 Controlled Gradient Descent

In this section, we extend our theoretical analysis of the controlled dynamical system back to the gradient descent algorithm. Recall that we are imposing the controller term on the second order

derivative $\frac{d^2\boldsymbol{\theta}'}{dt^2}$, which measures the acceleration of $\boldsymbol{\theta}$ with respect to the training time $t$. Therefore, we can easily recover the gradient $\frac{d\boldsymbol{\theta}'}{dt}$ of $\boldsymbol{\theta}$ by taking an integration on the second derivative $\frac{d^2\boldsymbol{\theta}'}{dt^2}$.

$$\frac{d\boldsymbol{\theta}'}{dt} = \int \frac{d^2\boldsymbol{\theta}'}{dt^2}\, dt = \int \frac{d^2\boldsymbol{\theta}}{dt^2} dt + \int u dt = \frac{d\boldsymbol{\theta}}{dt} - \frac{1}{2}K_1\boldsymbol{\theta}^2 - K_2\boldsymbol{\theta}, \tag{5}$$

where $\boldsymbol{\theta}^2$ is the element-wise square and $\boldsymbol{\theta}^2 := (\theta_1^2, \theta_2^2 ...., \theta_d^2)$. Notice that $\frac{d\boldsymbol{\theta}'}{dt}$ represents the gradient of $\boldsymbol{\theta}$ in continuous setting, in which we can extend to discrete gradient descent by considering it as $\frac{d\boldsymbol{\theta}'}{dt}|_{t=t}$, where $t$ is the current training time and we use $\boldsymbol{\theta}_t$ for evaluation. Specifically, we can apply modification into gradient computing process of gradient descent and propose our controlled gradient descent as:

---

**Algorithm 1:** Controlled Gradient Descent for Neural Network Training

**Input:** Neural network with parameters initialized as $\boldsymbol{\theta}_0$, learning rate $\eta > 0$, training data $\{(x_i, y_i)\}_{i=1}^N$, loss function $L(\boldsymbol{\theta}; x, y)$, maximum epochs $T$
**Output:** Trained network parameters $\boldsymbol{\theta}_T$
**for** $t = 0$ **to** $T - 1$ **do**
    **for** *each mini-batch* $\mathcal{B} \subset \{(x_i, y_i)\}$ **do**
        Compute gradient of loss: $\boldsymbol{g}_t = \frac{1}{|\mathcal{B}|}\sum_{(x_i,y_i)\in\mathcal{B}}(\nabla_{\boldsymbol{\theta}}L(\boldsymbol{\theta}_t; x_i, y_i) - K_1\boldsymbol{\theta}_t^2 - K_2\boldsymbol{\theta}_t)$;
        Update network parameters: $\boldsymbol{\theta}_{t+1} \leftarrow \boldsymbol{\theta}_t - \eta\boldsymbol{g}_t$;

**return** $\boldsymbol{\theta}_T$;

---

Intuitively, our controlled gradient descent can be considered as a gradient guidance toward the optimal equilibrium. In our theoretical analysis of GD, the stability of gradient descent is determined by the spectrum of the Hessian $H(\boldsymbol{\theta}^\star)$ at a local minimum $\boldsymbol{\theta}^\star$. Cohen et al. (2021) shows that the discrete system is stable only if the learning rate satisfies $\eta < 2/sharpness$. This criterion highlights the sharpness barrier that constrains the allowable learning rate.

We explain both stabilization and improved learning-rate tolerance from the eigenvalue shifting mechanism. In our controlled formulation, the update rule effectively replaces the original gradient $\nabla L(\boldsymbol{\theta})$ with a modified direction $\nabla L(\boldsymbol{\theta}) - K_1\boldsymbol{\theta}^2 - K_2\boldsymbol{\theta}$. The eigenvalues of the controlled Jacobian are therefore deriving from the shifted versions of those of $H(\boldsymbol{\theta}^\star)$. By choosing $K_1$ and $K_2$ following Definition 4, we guarantee local asymptotic stability of the continuous-time dynamics.

# 7 EXPERIMENTS

In this section, we empirically validate the effectiveness of our controlled gradient descent (CGD) algorithm on synthetic numerical cases. Our experimental design serves two complementary purposes. Even in the strongly convex and smooth case, vanilla GD may diverge when the learning rate is chosen outside the narrow stability region bounded by $2/sharpness$. Figure 1 highlights such instability in a toy quadratic problem, contrasting the divergent trajectory of GD with the stabilized dynamics achieved by CGD. This motivates a detailed study in the following subsections: (i) **Stability across various curvature regimes**, where we validate our CGD stabilize the training process regardless of curvatures; and (ii) **Stability under learning rates around the edge of stability**, where we show that CGD significantly enlarges the admissible step-size range compared to GD. Taken together, these experiments confirm our theoretical findings: CGD consistently stabilizes optimization across diverse curvature structures, demonstrates robustness to controller hyperparameters, and substantially improves tolerance to larger learning rates.

## 7.1 STABILITY FOR VARIOUS CURVATURE

We evaluate the stability of our controlled gradient descent comparing to GD across different curvature settings. Specifically, we consider three representative objective functions:

- Strongly convex ellipse: $L(\boldsymbol{\theta}) = 2\theta_1^2 + 0.5\theta_2^2$, initialized at $\boldsymbol{\theta}_0 = (2.0, 1.5)$ with $\eta = 0.5$.
- Strongly convex quartic: $L(\boldsymbol{\theta}) = \theta_1^4 + \theta_2^4$, initialized at $\boldsymbol{\theta}_0 = (1, 1)$ with $\eta = 0.5$.

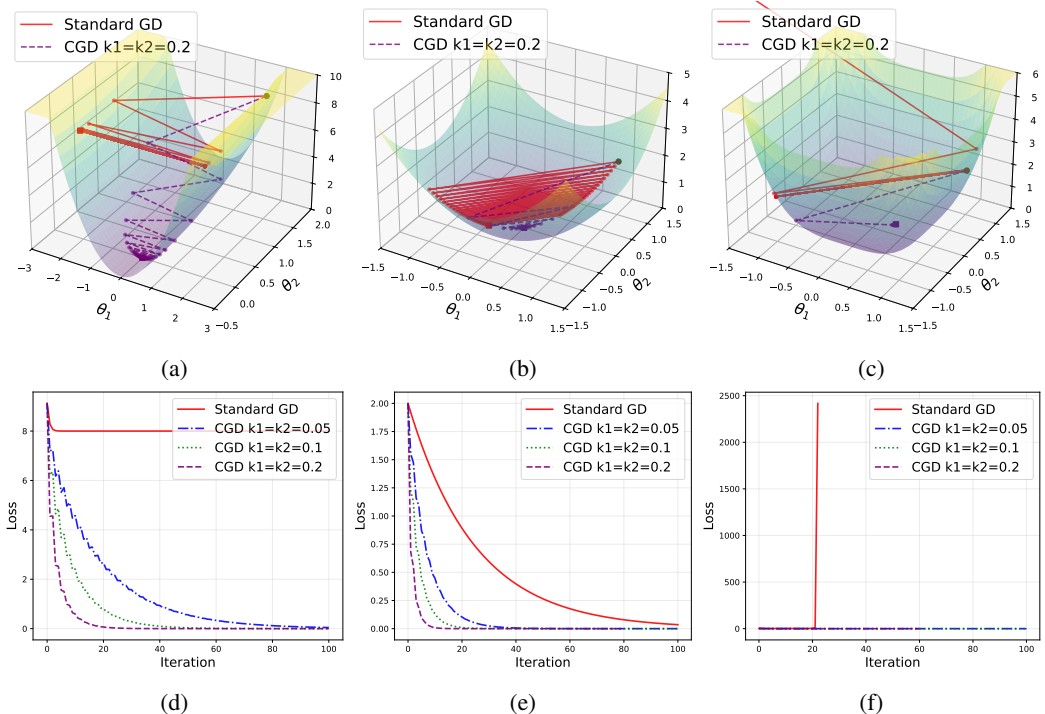

Figure 2: The optimization trajectories and training curves of GD and our controlled gradient descent on various curvatures. (a) and (d) is for strongly convex elliptical training loss $2\theta_1^2 + 0.5\theta_2^2$. (b) and (e) is for convex but not strongly convex sphere training loss $\theta_1^2 + \theta_2^2$. (c) and (f) is for strongly convex quartic training loss $\theta_1^4 + \theta_2^4$.

- Convex but not strongly convex sphere:$L(\boldsymbol{\theta}) = \theta_1^2 + \theta_2^2$, initialized at $\boldsymbol{\theta}_0 = (1, 1)$ with $\eta = 0.995$.

Figure 2 (a)–(c) show the optimization trajectories projected in three dimensions, while (d)–(f) depict the corresponding training curves. Across all cases, GD exhibits instability: oscillations on the ellipse, divergence on the quartic, and slow convergence or marginal instability on the sphere. In contrast, our contrast gradient descent consistently stabilizes the dynamics, ensuring convergence even when GD fails. This empirical evidence aligns with our theoretical analysis: the stability of GD is sensitive to curvature (both strong convexity and higher-order terms), while our control-theoretic modification guarantees asymptotic stability under all examined cases.

**Ablation on controller hyperparameters** We further investigate the sensitivity of controlled gradient descent to the choices of $K_1$ and $K_2$. In Figure 2 we set $K_1 = k_1 I, K_2 = k2I$, where we plot three curves for $k1 = k2 = 0.05, 0.1$ and $0.2$ respectively. We observe that our controlled gradident descent converges reliably regardless of the exact choice of hyperparameters. This indicates that the effectiveness of CGD does not hinge on fine-tuning $K_1$ and $K_2$, highlighting its robustness as a practical optimization method.

### 7.2 STABILITY FOR VARIOUS LEARNING RATE AROUND EOS

We analyze stability when the learning rate is close to the classical upper bound $\eta < 2/sharpness$. For the convex sphere loss $L(\boldsymbol{\theta}) = \theta_1^2 + \theta_2^2$ (sharpness = 2), we vary the learning rate around the theoretical threshold $\eta = 1$.

Figure 3 presents the loss curves under $\eta = 0.99, 1$ and $1.01$, respectively. We observe:

- For $\eta = 0.99$, GD converges slowly, while our method achieves faster and smoother convergence.

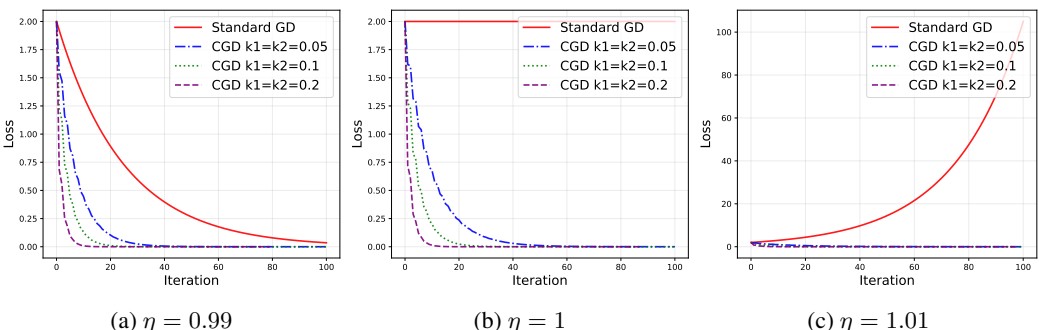

(a) $\eta = 0.99$         (b) $\eta = 1$         (c) $\eta = 1.01$

Figure 3: Comparison between the loss curves of GD and controlled gradient descent for strongly convex training loss $\theta_1^2 + \theta_2^2$ (sharpness = 2) for different learning rate around $2/sharpness$ under various settings of hyper-parameters for our controlled gradient descent

- At the critical point $\eta = 1.0$, GD fails to converge and oscillates around the sub-optimum, while our method maintains stability.
- For $\eta = 1.01$, GD diverges, but our method continues to converge reliably.

These results demonstrate that our controlled gradient descent remains stable beyond the edge of stability, validating its robustness with respect to learning rate selection.

## 8 CONCLUSION AND DISCUSSION

In this paper, we propose a controlled gradient descent method using control theory to stabilize the training dynamics of GD. We formulate GD as a second-order dynamical system and use this perspective to analyze its stability. Through this reformulation, we show that GD can diverge even when the learning rate satisfies the classical bound, highlighting fundamental limitations of existing stability analyses. We further characterize how stability behaviors differ under various curvature conditions, demonstrating that convergence cannot be guaranteed solely by bounding the learning rate. To address these issues, we introduce a controller that regulates the eigen-structure of the training loss Hessian. We prove that this controller guarantees local asymptotic stability under general curvature settings and interpret it as a gradient guidance term augmenting the original update rule. This control-theoretic lens opens a pathway for systematically designing stabilized variants of gradient descent that remain effective in highly non-convex or non-smooth landscapes. Empirical evaluations on synthetic problems confirm our controlled gradient descent improves stability, tolerates larger learning rates, and converges more reliably than standard GD.

**Limitations and Future Directions:** Our analysis focuses on the continuous-time formulation of gradient descent, where the learning rate is assumed to be sufficiently small so that the discrete updates approximate the gradient flow. Within this setting, we show that gradient descent can still diverge under various curvature conditions, revealing instability that persists even in the idealized continuous case. However, a gap remains between continuous-time differential equations and the actual discrete gradient descent updates. This gap represents a limitation of our current analysis, as discretization effects may introduce additional sources of instability or alter the stability thresholds we derive. Future work includes conduction stability analysis directly in discrete setting. Extending the controller design to stochastic optimization, adaptive learning-rate methods, and large-scale non-convex landscapes also represents an exciting direction for building more robust training algorithms.

## 9 BROADER IMPACTS AND LLM USAGE

This work is primarily theoretical and focuses on the stability analysis of gradient descent from a control-theoretic perspective. Its contributions lie in advancing the understanding of optimization dynamics and in proposing more stable training methods. Any broader societal consequences would only arise indirectly through downstream applications of deep learning, which fall outside the scope of this study. We have used LLM to polish writing for this paper.

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

## A    APPENDIX A: PROOF OF LEMMA 1

Let $L : \mathbb{R}^n \to \mathbb{R}$ be a twice continuously differentiable function. Then, $L$ is strongly convex if and only if there exists a constant $m > 0$ such that the Hessian satisfies

$$\nabla^2 L(\boldsymbol{\theta}) \succeq mI \quad \text{for all } \boldsymbol{\theta} \in \mathbb{R}^n.$$

Equivalently, $L$ is strongly convex if and only if $\nabla^2 L(\boldsymbol{\theta})$ is positive definite for all $\boldsymbol{\theta} \in \mathbb{R}^n$.

*Proof.* We prove both directions.

($\Rightarrow$) **Strong convexity implies positive definiteness.**    Assume $L$ is $m$-strongly convex. By definition, for all $x, y \in \mathbb{R}^n$,

$$L(y) \geq L(x) + \nabla L(x)^\top (y - x) + \frac{m}{2} \|y - x\|^2.$$

Since $L$ is twice differentiable, we can apply the second-order Taylor expansion around $x$:

$$L(y) = L(x) + \nabla L(x)^\top (y - x) + \frac{1}{2}(y - x)^\top \nabla^2 L(\xi)(y - x),$$

for some $\xi$ on the line segment between $x$ and $y$. Comparing with the strong convexity inequality, we obtain:

$$\frac{1}{2}(y - x)^\top \nabla^2 L(\xi)(y - x) \geq \frac{m}{2} \|y - x\|^2,$$

which implies:

$$(z)^\top \nabla^2 L(\xi)z \geq m\|z\|^2, \quad \forall z = y - x \in \mathbb{R}^n.$$

Therefore, $\nabla^2 L(\xi) \succeq mI$, which means $\nabla^2 L(\boldsymbol{\theta}) \succ 0$ for all $\boldsymbol{\theta} \in \mathbb{R}^n$.

($\Leftarrow$) **Positive definiteness implies strong convexity.**    Assume $\nabla^2 L(\boldsymbol{\theta}) \succeq mI$ for some $m > 0$, and all $\boldsymbol{\theta} \in \mathbb{R}^n$. Using Taylor's expansion as above, we again write for all $x, y \in \mathbb{R}^n$:

$$L(y) = L(x) + \nabla L(x)^\top (y - x) + \frac{1}{2}(y - x)^\top \nabla^2 L(\xi)(y - x),$$

for some $\xi$ on the segment between $x$ and $y$. Then:

$$(y - x)^\top \nabla^2 L(\xi)(y - x) \geq m\|y - x\|^2,$$

and so:

$$L(y) \geq L(x) + \nabla L(x)^\top (y - x) + \frac{m}{2} \|y - x\|^2,$$

which is the definition of $m$-strong convexity. Hence, $L$ is strongly convex.    $\square$

## B    APPENDIX B: PROOF OF LEMMA 2

*Proof.* We prove both directions.

($\Rightarrow$) **Convexity implies positive semidefiniteness.**    Assume $L$ is convex. By the definition of convexity, for all $x, y \in \mathbb{R}^n$,

$$L(y) \geq L(x) + \nabla L(x)^\top (y - x).$$

Using the second-order Taylor expansion at $x$, we have:

$$L(y) = L(x) + \nabla L(x)^\top (y - x) + \frac{1}{2}(y - x)^\top \nabla^2 L(\xi)(y - x),$$

for some $\xi$ on the segment joining $x$ and $y$. Comparing this with the convexity inequality gives:

$$\frac{1}{2}(y - x)^\top \nabla^2 L(\xi)(y - x) \geq 0,$$

which implies:

$$z^\top \nabla^2 L(\xi)z \geq 0 \quad \text{for all } z = y - x \in \mathbb{R}^n.$$

Thus, $\nabla^2 L(\xi) \succeq 0$, and since $\xi$ is arbitrary, the Hessian is positive semidefinite everywhere.

($\Leftarrow$) **Positive semidefiniteness implies convexity.** Assume $\nabla^2 L(\boldsymbol{\theta}) \succeq 0$ for all $\boldsymbol{\theta} \in \mathbb{R}^n$. Let $x, y \in \mathbb{R}^n$, and consider:

$$\phi(t) = L(x + t(y - x)), \quad t \in [0, 1].$$

Then:

$$\phi''(t) = (y - x)^\top \nabla^2 L(x + t(y - x))(y - x) \geq 0.$$

So $\phi$ is a convex function on $[0, 1]$, and:

$$\phi(1) \geq \phi(0) + \phi'(0) = L(x) + \nabla L(x)^\top (y - x).$$

This gives the first-order convexity condition, and hence $L$ is convex. $\qquad\square$

## C  APPENDIX C: PROOF OF LEMMA 3

Let $L : \mathbb{R}^n \to \mathbb{R}$ be twice continuously differentiable. Then

$$L \text{ is concave} \quad \Longleftrightarrow \quad \nabla^2 L(\boldsymbol{\theta}) \preceq 0 \text{ for all } \boldsymbol{\theta} \in \mathbb{R}^n,$$

equivalently, every eigenvalue of $H(\boldsymbol{\theta}) = \nabla^2 L(\boldsymbol{\theta})$ satisfies $\lambda_i(H(\boldsymbol{\theta})) \leq 0$.

*Proof.* We prove both directions.

($\Rightarrow$) **Concavity implies negative semidefiniteness.** Assume $L$ is concave. Fix $\boldsymbol{\theta} \in \mathbb{R}^n$ and $\boldsymbol{h} \in \mathbb{R}^n$ and define the univariate function

$$\varphi(t) := L(\boldsymbol{\theta} + t\boldsymbol{h}), \quad t \in \mathbb{R}.$$

Concavity of $L$ implies $\varphi$ is concave on $\mathbb{R}$, hence $\varphi''(t) \leq 0$ for all $t$. By the chain rule,

$$\varphi''(t) = \boldsymbol{h}^\top \nabla^2 L(\boldsymbol{\theta} + t\boldsymbol{h})\, \boldsymbol{h}.$$

Evaluating at any $t$ (in particular $t = 0$) yields

$$\boldsymbol{h}^\top \nabla^2 L(\boldsymbol{\theta})\, \boldsymbol{h} \leq 0 \quad \text{for all } \boldsymbol{h} \in \mathbb{R}^n,$$

which is exactly $\nabla^2 L(\boldsymbol{\theta}) \preceq 0$.

($\Leftarrow$) **Negative semidefiniteness implies concavity.** Assume $\nabla^2 L(\boldsymbol{\theta}) \preceq 0$ for all $\boldsymbol{\theta}$. Fix $\boldsymbol{\theta}, \boldsymbol{h}$ and define $\varphi(t) := L(\boldsymbol{\theta} + t\boldsymbol{h})$. Then

$$\varphi''(t) = \boldsymbol{h}^\top \nabla^2 L(\boldsymbol{\theta} + t\boldsymbol{h})\, \boldsymbol{h} \leq 0 \quad \text{for all } t.$$

Thus $\varphi$ is concave on $\mathbb{R}$. Since $L$ is concave along every line in $\mathbb{R}^n$, it is concave on $\mathbb{R}^n$. $\qquad\square$

**Eigenvalue Corollary.** From $\nabla^2 L(\boldsymbol{\theta}) \preceq 0$ it follows that all eigenvalues of $H(\boldsymbol{\theta})$ are nonpositive: $\lambda_i(H(\boldsymbol{\theta})) \leq 0$ for $i = 1, \ldots, n$.

