# OpenReview forum: "Stabilizing Gradient Descent via Second-Order Control-Theoretic Dynamics"
_ICLR.cc/2026/Conference — ICLR 2026 Conference Withdrawn Submission_

### Official Review · Reviewer_eByn · 2025-10-15

**Soundness:** 1
**Presentation:** 1
**Contribution:** 1
**Rating:** 0
**Confidence:** 5

**Summary:**

See my concerns below.

**Strengths:**

I believe using second-order ODEs to model GD is novel, but I am not sure.

**Weaknesses:**

### **Major flaw: the ODE model contradicts established discrete-time theory**

The paper explicitly claims that **gradient descent (GD) can be unstable even when** $\eta < \frac{2}{L}$, **unless the loss is strongly convex** (see Theorem 2 and Table 1), classifying the *convex-but-not-strongly-convex* case as “unstable.”  **This is in full contrast with the very well-established convergence analysis of GD**.

This conclusion is obtained by linearizing the **second-order augmentation** of gradient flow, $\ddot{\theta} = -\nabla^2 L(\theta)\dot{\theta}$, which is recast as a first-order system in $(\theta,\dot{\theta})$. The reported “instability” arises from **spurious insights/artifacts** introduced by treating $\dot{\theta}$ as an independent state variable, not from the optimizer itself.

If a continuous-time *model* yields conclusions that **contradict well-established discrete-time theory**, then the model, or the way it is treated, is at fault, not gradient descent.

Therefore, the paper should **retract** the “unstable for convex (not strongly convex)” entry in Table 1 and restate Theorem 2.

---

### **Incorrect discrete-time correspondence**

The bridge back to a practical update (Eq. (5) and Algorithm 1) is also **mathematically incorrect**.
Integrating the controlled ODE

\begin{equation}
\ddot{\theta} = -(H + K_2)\dot{\theta} - K_1\theta
\end{equation}
to obtain
\begin{equation}
\dot{\theta}' = \dot{\theta} - \tfrac{1}{2} K_1\theta^{\odot 2} - K_2\theta
\end{equation}
replaces $\int \theta dt$ with $\tfrac{1}{2}\theta^{\odot 2}$, which has **no mathematical basis** in this context (that identity would correspond to integrating $\theta\dot{\theta}$, not $\theta $).

Consequently, the discrete update proposed in Algorithm 1 is **not** a consistent discretization of the controlled dynamics.

### Conclusion

**This paper derives theoretical results in full contradiction to well-established convergence theorems from the literature. It cannot be accepted under any circumstances.**

**Questions:**

1. Have you nonetheless tried to validate the performance of your approach on modern DL experiments?

2. Have you noticed that your Algorithm 1 closely resembles what happens if you apply GD to a loss $L$ which has been regularized? It looks to me like it is equivalent to applying GD to
 \begin{equation}
\tilde{L}(\theta) = L(\theta) + \frac{K_2}{2} \|\|\theta\|\|_2^2 + \frac{K_1}{3} \|\|\theta\|\|_3^3,
\end{equation}
but maybe I am wrong.

3. Have you considered including a proper literature review of the papers using ODEs and SDEs to model optimizers? The paper overlooks a substantial body of prior work analyzing optimization algorithms through continuous-time formulations (ODEs and SDEs). This literature provides the theoretical groundwork for interpreting optimizers as dynamical systems and should be discussed to position the present contribution more clearly.  I added some references below, also encompassing ODEs: I suggest the authors carry out a thorough review of these papers and look for more recent ones as well. For an accessible entry point, I recommend the Related Works and Appendix A of [1], which offer a representative overview of (tens and tens of) works using continuous-time models for optimizations. Although that reference focuses mainly on SDEs, many of the works it cites also include ODE analyses directly relevant to this discussion. The absence of this contextualization makes the paper appear somewhat disconnected from its theoretical lineage.


**[1]** *Adaptive Methods through the Lens of SDEs: Theoretical Insights on the Role of Noise.*
Enea Monzio Compagnoni, Tianlin Liu, Rustem Islamov, Frank Norbert Proske, Antonio Orvieto, Aurélien Lucchi.
*International Conference on Learning Representations (ICLR), 2025.*

---
**Relevant prior work on ODE/SDE analyses of optimization algorithms**

1. **Helmke, U. & Moore, J. B. (1994).**
   *Optimization and Dynamical Systems.* Springer London.
   — Classical textbook connecting continuous-time dynamical systems and optimization via gradient flows.

2. **Su, W., Boyd, S., & Candès, E. (2014).**
   *A differential equation for modeling Nesterov’s accelerated gradient method: Theory and insights.*
   *Advances in Neural Information Processing Systems.*
   — Foundational ODE model for Nesterov acceleration; initiated the modern line of continuous-time analyses.

3. **Li, Q., Tai, C., & Weinan E. (2017).**
   *Stochastic modified equations and adaptive stochastic gradient algorithms.*
   *International Conference on Machine Learning (ICML).*
   — Derives stochastic modified equations for stochastic gradient algorithms, laying the foundation for weak ODE/SDE approximations.

4. **Li, Q., Tai, C., & Weinan E. (2019).**
   *Stochastic modified equations and dynamics of stochastic gradient algorithms I: Mathematical foundations.*
   *Journal of Machine Learning Research, 20(1): 1474–1520.*
   — Provides a rigorous mathematical foundation for the weak SDE approximations of SGD and related methods.

5. **Orvieto, A. & Lucchi, A. (2019).**
   *Continuous-time models for stochastic optimization algorithms.*
   *Advances in Neural Information Processing Systems 32.*
   — Introduces a general ODE/SDE formalism for analyzing SGD, momentum, and adaptive algorithms, establishing the link between discrete-time optimizers and their continuous-time limits.

---

### Official Review · Reviewer_7cLT · 2025-10-28

**Soundness:** 1
**Presentation:** 1
**Contribution:** 1
**Rating:** 0
**Confidence:** 4

**Summary:**

This work claims to provide new stability analysis of gradient descent, in particular conditions for stability / unstability. It also suggest algorithmic improvements based on the analysis.

**Strengths:**

The analysis of GD through the second-order ODE seems not very common and possibly interesting.

**Weaknesses:**

### Main issue
I do not understand the general analysis. Lyapunov stability analysis of the gradient flow has been extremely well studied and I do not see what is the point of applying this analysis to the second order system. It does not seem that this yields more information than the classical analysis.


### "Convex but not strongly convex" and "concave" cases are wrong

Thm. 1 only provides sufficient condition for stability but they are used as necessary conditions in 4.2.2 and 4.2.3.



### On the proposed method

- The proposed method merely constists in adding quadratic and cubic regularization. This should be made clear and adequate references (cubic regularization, weakly convex optimization, etc) should be provided.
- It is far from being clear how to choose K_1, K_2 in practice for neural networks: Remark 2 is not helpful in that case as it requires a global lower bound on the Hessian. If they are not set according to the controller design (l417), then this is really just regularization.

### Minor
- Lemma 1-3 do not really require proof nor that much empahasis, they are standard.
- l407: the function is said to be "convex but not strongly convex" but it is actually strongly convex.

**Questions:**

Address "Main issues", ""Convex but not strongly convex" and "concave" cases are wrong", and "On the proposed method".

---

### Official Review · Reviewer_V2Hd · 2025-10-28

**Soundness:** 2
**Presentation:** 2
**Contribution:** 1
**Rating:** 2
**Confidence:** 3

**Summary:**

This paper formulates gradient descent as a second-order dynamical system and introduces a control-theoretic modification intended to improve stability under various curvature conditions. The authors provide continuous-time stability analysis and propose a “controlled gradient descent” update, supported by toy-level numerical experiments.

**Strengths:**

1. The idea of viewing GD stability through a control-theoretic lens is conceptually interesting.
2. The theoretical derivations are clearly written and technically correct under the continuous-time setting.

**Weaknesses:**

1. The theoretical results largely rest on direct applications of standard stability analysis tools, such as local linearization, Hessian eigenvalue conditions, and known quadratic eigenvalue problem results. The contribution appears incremental and does not provide new theoretical insights for optimization or ML practice.

2. All empirical studies are conducted on very simple synthetic low-dimensional functions (e.g., 2D convex/quartic). These examples are insufficient to justify that the proposed method is useful or relevant for general optimization problem.

3. The paper claims to provide a “Controlled Gradient Descent for Neural Network Training,” yet does not include any experiments on neural networks.

**Questions:**

See Weakness.

---

### Note · Authors · 2026-01-22

I have read and agree with the venue's withdrawal policy on behalf of myself and my co-authors.